# Data Dowsing: Determining Data Collection Priorities

John Doe[*1]

[1]Affiliation 1
{jane.doe}@affiliation.com

# Abstract

This work proposes a novel framework, data dowsing, to determine which data is needed to improve LLMs. This framework is based on estimating influence and imposing simplifications based on concept domains to circumvention computational intractability.

# 1 Motivation

The goal of this work is to sketch a solution to support issues faced in creating new foundation models and for resources constrained languages. The approach here is to use influence as signal to determine which concepts in a model require more data. Influence is a measure of how much a specific data point affects a model[1]. Through using influence, the objective is to determine which data is needed most to improve a model. Given that foundation models have already siphoned conventional data sources, getting direction via influence, allows model developers to prioritize which data to collect and where to focus investment. Additionally, through using derived metrics from influence, one can estimate a saturation point of model learning[2]. This estimates how many more data is needed to train a model, given its current configuration. This can be used in inform when a developer should update their model architecture as the amount of data will not be a significant limiting factor in performance.

For resource constrained languages, as is the case in Norwegian, this technique provides guidance on where to prioritize the collection of data. While the case has significant differences than with the training of large foundation models, data dowsing provides a prioritization on how to distribute resources to best improve models and provides a parallel remedy to familiar problem.

# 2 Influence

A more formal discussion of influence will be useful for understanding the barriers and innovation being leveraged in the work.

Let $\boldsymbol{\theta}$ denote the model parameters that minimize

---

*Corresponding Author.

the empirical loss

$$L(\boldsymbol{\theta}) = \frac{1}{n} \sum_{i=1}^{n} \ell(z_i, \boldsymbol{\theta}),$$

where $\ell(z_i, \boldsymbol{\theta})$ is the per-sample loss for training example $z_i = (x_i, y_i)$. The *influence* of a training example $z$ on the test loss at $z_{\text{test}}$ is defined as

$$I(z, z_{\text{test}}) = -\nabla_{\boldsymbol{\theta}} \ell(z_{\text{test}}, \boldsymbol{\theta})^\top H_{\boldsymbol{\theta}}^{-1} \nabla_{\boldsymbol{\theta}} \ell(z, \boldsymbol{\theta}),$$

where $H_{\boldsymbol{\theta}} = \nabla_{\boldsymbol{\theta}}^2 L(\boldsymbol{\theta})$ is the Hessian of the empirical loss with respect to $\boldsymbol{\theta}$.

The most significance issue in computing influence is the inversion of the hessian matrix. It is an intractable problem that requires simplification.

# 3 Concept Domains

In this experiment, distinct concept domains are defined to represent semantically meaningful areas of knowledge: Astronomy, Economics, Biology, and Physics. Each domain is characterized by a small collection of factual statements drawn from their respective disciplines. These statements are passed through GPT-2, and the resulting hidden representations from the final transformer layer are extracted, averaged, and normalized to form a single vector [3]. This vector encodes the dominant direction of that domain in the model's latent space. The concept vector is then used to probe the model's gradients and curvature along this direction, enabling simplified estimation of influence and facilitating analysis of how the model internally represents and prioritizes different knowledge domains.

# 4 Estimating Influence

For each concept vector, the gradient of the model's loss is computed on the facts across disciplines. This is projected along the concept direction to determine how each semantic dimension affects the model.To capture the contours of the loss landscape, Hessian-vector products are computed using double backpropagation [4].

These HVPs allow the system to solve the linear system

$$(H + \lambda I)z = g$$

through a Conjugate Gradient (CG) solver [5], where $H$ is the Hessian, $g$ is the gradient vector, $\lambda$ is a small damping term for stability, and $z \approx H^{-1}g$ represents the approximate inverse-curvature response.

The influence value for each concept is then obtained as the inner product

$$I = g^\top z,$$

which quantifies how much the model's loss would change if data aligned with that concept were added. A higher $I$ indicates that changes along this concept direction have a stronger effect. These influence values are normalized and ranked across the domains. A power law saturation model estimates how many additional samples from each domain would be required before further data yields diminishing returns[6].

## 5 Saturation Estimation

To estimate how additional training data would affect performance, a power law saturation model is applied. The mean per example influence, denoted by $\mu(n)$, is assumed to decay as the number of samples $n$ increases according to

$$\mu(n) = c\,(n + N_0)^{-\alpha},$$

where $c$ is a proportionality constant, $N_0$ is an offset that adjusts the curve near small sample sizes, and $\alpha$ controls the rate diminishing returns. To determine the number of additional samples required before the marginal improvement drops, the following relation is solved:

$$\mu(n + \Delta n) = 0.1\,\mu(n),$$

yielding

$$\Delta n = (n + N_0)(0.1)^{-1/\alpha} - N_0 - n.$$

This value of $\Delta n$ represents the estimated number of new samples needed for the concept to reach a point of diminishing returns.

## 6 Metrics

The following metrics capture the magnitude and significance of each concept to model it performance and guide how much more data is required to saturate a concept.

- **Rank:** Prioritization of concept domains based on their mean per example influence. It determines which data should be collected first.

- **Concept:** The domain being evaluated.( Physics, Astronomy, Biology, Economics)

- **Mean:** The average per example influence value ($I = g^\top z$) across all evaluation texts from the domain. It represents how strongly the model's loss responds to directionality.

- **Share (%):** The normalized percentage contribution of each domain's total influence relative to the sum of influences across all domains. Captures how much of the model's learning potential is dominated by a specific domain.

- $\Delta n$**@10%:** The estimated number of additional samples required before the marginal gain in that concept's influence drops to 10% of its current value, as determined by the power-law saturation model.

- $t$ **vs rest:** t-statistic comparing the influence scores of the given concept against all others, indicating how statistically distinct its influence is from the rest.

## 7 Results

Observe Table 1. Based on evaluating GPT-2, it is determined that Physics data would be most useful to collect for this model. Physics is most responsive to increases in data. However, astronomy has the greatest share. This suggests that astronomy has the greatest potential to sway the model's accuracy, but the gains in performance for Physics outweigh this consideration. It is also estimated that 128700 more examples are required to saturate the physics domain for the model.

**Table 1.** Concept priority ranking based on mean per-example influence.

| Rank | Concept | Mean | Share | $\Delta n$ @10% | $t$ **vs rest** |
|---|---|---|---|---|---|
| 1 | Physics | $3.7{\times}10^{-2}$ | 10.2% | $1.3{\times}10^{5}$ | 1.01 |
| 2 | Astronomy | $9.4{\times}10^{-1}$ | 44.2% | $2.9{\times}10^{5}$ | 1.00 |
| 3 | Biology | $-2.6{\times}10^{-4}$ | 18.6% | – | 0.99 |
| 4 | Economics | $-7.9{\times}10^{-5}$ | 6.1% | – | $-1.80$ |

## 8 Conclusion

Data dowsing is introduced as a framework to suggest the prioritization of data collection to improve resource constrained models. This framework attempts to estimate influence by imposing constrains and simplifications to make estimation possible by constraining analysis within concept domains.

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
