# OpenReview forum: "Data Dowsing: Determining Data Collection Priorities"
_NLDL.org/2026/Abstracts_Track — NLDL 2026 Abstracts_

### Official Review · Reviewer_289V · 2025-11-03

**Soundness:** 3
**Correctness:** 3
**Rating:** 4
**Confidence:** 4

**Summary:**

The objective is to determine how to reduce the data fed to an LLM.

**Strengths:**

Needed by the field.

**Weaknesses:**

This is a 2-line abstract; I cannot evaluate their weaknesses.

---

### Official Review · Reviewer_cGwt · 2025-11-03

**Soundness:** 4
**Correctness:** 3
**Rating:** 5
**Confidence:** 4

**Summary:**

This work proposes a method to select data points needed for LLMs to improve their performance. The main idea is to estimate influence by using concept domains. They estimate the influence by using hessian matrix, and provide an approach to make the matrix inversion computationally tractable.

**Strengths:**

The work is really interesting!

* Well written and clear presentation of the idea
* The experiment and explanation of mathematical operators is clear and interesting

**Weaknesses:**

* An illustration would make it much more interesting
* HVP, I got that this is the Hessian Vector Products, but you need to mention in the abstract what are you abbreviating

* For future work, it would be really important to answer the following:

1. How this affects the catastrophic forgetting, and how is this related to continual learning?
2. How it would affect not only the predictive performance, but also hallucinations, calibration, and uncertainty of the model?
3. Could we extend also to images? measuring the influence of data point on the test point? and how to make it much more tractable. Cause images contain much more information, and the visual concepts are not clearly stated compared to textual information where you have "Factual statements"
4. How is this related to meta learning?
5. It would be also nice to check and combine the idea with the following line of research: https://arxiv.org/pdf/2410.08020

---

### Official Review · Reviewer_mnYB · 2025-11-04

**Soundness:** 3
**Correctness:** 3
**Rating:** 4
**Confidence:** 2

**Summary:**

Tha authors propose a framework, data dowsing, to select data to improve LLMs.
This is particularly useful in languages with fewer resources available than English.
The framework is based off influence, which is a measure of how much a data point affects a model.

**Strengths:**

- The idea of “data dowsing” is engaging and original. It provides a nice idea for data prioritisation and ties nicely to real challenges in improving LLMs and low-resource languages.
- The use of influence functions and power-law saturation modeling shows good knowledge of  theoretical tools, giving the work a solid conceptual foundation even in its preliminary form.

**Weaknesses:**

- The abstract describes procedures and equations in detail but provides little discussion of experimental results supporting the framework’s usefulness beyond a single table.
- The experiments are carried out on GPT-2, which is now a bit outdated; the work would be stronger if the experiments were carried out on newer classes of models (Llama, or GPT-3-class systems)
- Some sentences are grammatically awkward (e.g., “to circumvention computational intractability”)

---

### Decision · Program_Chairs · 2025-11-05

**Decision:**

Accept

**Comment:**

The abstract is of interest to the community and should be presented at the conference.